# Does a degree in medicine or a specialist programme or socioeconomic status advance the career of general practitioners in primary healthcare?

Dandan Zheng[1,2], Gazi Mahabubul Alam [3]*, Karima Bashir[4,5], Norlizah Che Hassan[2], Miao Lei[6]

**1** Department of Science and Technology, Guizhou Medical University, Guiyang, Guizhou, China,
**2** Department of Foundation of Education, Faculty of Educational Studies, University Putra Malaysia, Serdang, Malaysia, **3** School of Education, Faculty of Social Sciences and Leisure Management, Taylor's University, Subang Jaya, Malaysia, **4** Department of Educational Foundation, Faculty of Education and Extension Services, Usmanu Danfodiyo University, Sokoto, Nigeria, **5** Department of Education, Faculty of Education, Kebbi State University of Science and Technology, Aliero, Kebbi State, Nigeria, **6** Student Affairs Division, Yancheng Teachers University, Yancheng, Jiangsu, China

* gazimalamb@yahoo.com

## Abstract

Extensive studies have investigated possible mechanism to improve the career path of general practitioners (GPs) in order to ensure the primary healthcare system (PHC) functions properly. Nevertheless, this problem persists in many countries and it is aggravated by preferential policy. Against this background, the current study examined whether the diverse groups of GPs employed by the PHC system help to develop a sustainable PHC system or create a dysfunctional one. This quantitative cross-sectional survey sampled 12,767 graduates from 13 medical schools in China. Secondary data was collected on socioeconomic status (SES), career advancement, educational qualification, demographic characteristics and graduates' employment information. The collected data was analyzed using group regression and mediation analysis. Results indicated that, among the 1,769 clinical physicians working in PHC institutions, the ratio of specialist to specialists transferred general practitioners and full general practitioners is 3:2:1, indicating a horizontal mismatch. Furthermore, medical graduates' degree program type significantly relates to their career advancement and SES mediated the link between medical graduates' degree program type and career advancement with an indirect effect: Internal Medicine SP-GP→SES→CA ($\beta = 0.040$, $P < 0.001$); Obstetrics and Gynecology SP-GP→SES→CA ($\beta = 0.026$, $P < 0.001$); Emergency Medicine SP-GP→SES→CA ($\beta = 0.021$, $P < 0.001$); Paediatrics SP-GP→SES→CA ($\beta = 0.024$, $P < 0.001$). This study contributes to discourse on SES and medical education by highlighting how medical education has evolved into a money-driven program in which students' SES influences the program they select.

**Data availability statement:** All data are in the manuscript and/or Supporting Information files.

**Funding:** This research was supported by the Philosophy and Social Sciences Planning Project of Guizhou Province, 2023 (Project No. 23GZQN73). The funders had no role in study design, data collection and analysis, decision to publish, or preparation of the manuscript.

**Competing interests:** The authors declare that they have no competing interests.

## Introduction

Primary health care (PHC) systems and their criteria are shaped by social and historical practices [1]. However, workforce allocation and planning in the healthcare sector are mainly driven by market-oriented reforms and commercialization [2]. In these circumstances, general practitioners (GP) are considered pivotal within the health care system. GPs are custodians of the health system. They are responsible for the types of care that are practised in PHC. They guide and manage patients' health problems and ensure the connection between different services, by providing comprehensive and patient-centered medical care [3]. Likewise, they provide diagnostic support for complex cases, propose higher-level treatment plans, or referrals to specialists.

However, since the end of the 19th century, the number of GPs in the workforce has gradually declined, raising concerns about the sustainability of PHC [4]. Fallows [5] explained that the rapid digitalization, changes in people's dietary habits, urbanized lifestyles, and the continuous eruption of contagious diseases has pushed the need to prioritize production of specialist practitioner. This has further contributed to the decline of the development and employment of GPs. Coupled by modernization and capitalism, GPs have had to work in conditions of insufficient funding and limited professional recognition [6].

### Research gap and scope

Numerous specific initiatives such as the salary base reforms and performance bonuses in Australia and Canada [7,8], and the billing autonomy policy in Germany [9], and many other medical reforms have been implemented to address the shortage of GPs. However, in developing countries, the challenge persists as governments have failed to provide a common solution that addresses the primary structural challenges in the PHCs. Specifically, for China which is the subject of this paper, an especial policy is implemented which allows clinical physicians who are specialized in the treatment of specific diseases to transfer to general practice, while general practitioners are not allowed to move to specialist practice. This preferential policy has exacerbated the deeply rooted challenges of GPs and PHCs in China and caused employment mismatch.

However, this structural problem has not been discussed in depth in existing studies. The available literature has explored feasible strategies to improve the working conditions of GPs [10,11]. Likewise, Shi et al. [11] and Kongstad et al. [12] concluded that in recent years, a large number of specialists may have entered the GP vocation through the transfer mechanism, and this trend is worthy of further investigation. In addition, Ogden et al. [13] explored the 'rural pipeline' program; Porkodi et al. [14] and Marchand and Peckham [15] discussed strategies for prioritizing career advancement paths and improving the work environment. Similarly, Moad et al. [16] examined the relationship between the overall performance of GPs and their SES. However, these studies did not fundamentally address the impact of this policy on "labor dynamic imbalance" caused by the entry of specialists into PHC.

Moreover, Khan et al. [17] and Puddey et al. [18] have noted that students from lower SES backgrounds are more likely to choose the vocation of GP as a career path. Meanwhile those from higher SES backgrounds are more likely to choose specialist training which has advantageous career prospects. Based on this background, a fundamental question that needs to be answered is: whether in the career development of GPs, is it a medical degree, specialist training, or SES that plays a dominant role? This question serves as the motivation for this comparative study.

## Background on the Chinese medical training system

Prior to the 1950s, the Chinese medical education system was not formalized by the government as one national health system. Rural communities overwhelmingly relied on short-term trained "barefoot doctors" to meet the basic needs of people who were essentially poor and did not have the means to pay the medical profession. Later, these barefoot doctors gradually developed into a cooperative medical system, and it became the core pillar of rural medical services, although they received mainly on-the-job training. In contrast, urban areas have established a structured three-level prevention network with more complete equipment and higher levels of medical service capacity [19]. Thus, China operates its medical system based on an urban-rural dual-track pattern [20].

By the late 1990s, the barefoot doctors co-operations were formalized and improved mainly through capacity building and the establishment of general practice as a clinical discipline [19]. This gave birth to the modern GPs. Meanwhile, the urban health system was developed as the tertiary health system (SPs) and both GPs and SPs were trained and certified through medical training colleges. However, GPs (often associated with lower SES) still face many challenges in practice, such as the lack of a clear role definition [21]. In addition, because patients can directly seek medical treatment in tertiary hospitals without referral from primary care, patients with mild symptoms generally bypass PHC centers leading to further marginalizing GPs [22]. Moreover, China's decentralization of financial responsibility for PHC institutions to local governments has further amplified resource challenges for GP and PHC institutions.

The Chinese medical training system for GPs currently has three main pathways as follows: the "5+3" standardized residency training (SRT) (five years of undergraduate clinical medicine education; this is followed by either three years of SRT or three years of a Master's program in clinical medicine), GP transition training, and the on-the-job training [11]. Despite the increase in the number of GPs and PHC institutions over the years, the imbalance in the PHC workforce has continued (see Figs 1,2, and 3, respectively).

Based on what is presented in the figures, it is clear that the policy of promoting specialists to work as GPs has helped to alleviate the scarcity of GPs in China's PHC institutions. It has, however, caused employment mismatches.

## Research aim, questions and hypothesis

Career advancement is a multifaceted term, encompassing indicators like salary levels or growth, level of management status achieved, and promotion opportunities [23–25]. It refers to the systematic progression, clinical expertise, professional responsibility and leadership roles of general practitioners shaped by the evolving needs of the PHC system. Its dimensions include both vertical development, such as promotion or clinical grade advancement, and horizontal development, such as skill diversification and role expansion [25]. In this study, these indicators will be used as the main measures to assess the career advancement of GPs.

It is worth noting that prior studies have suggested inadequate GP career advancement is the primary cause for weak PHC institutions [26,27]. Hence, numerous studies have proffered various solutions such as financial incentives and expanding GP roles. However, no solution has led to the required outcome [28]. Interestingly, Dowell et al. [29] in their studies assert that GPs with higher SES are able to work better. This garnered the attention researchers to explore whether SES plays a dominant role in determining the career advancement of GPs. Against this background, the current research challenged conventional works that studied GP as a homogeneous group as presented in Fig 4, potentially

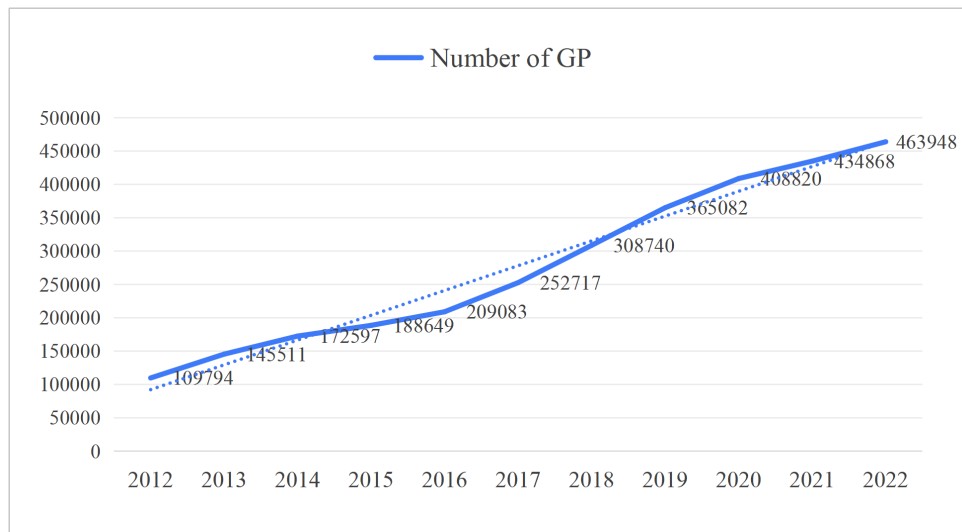

**Fig 1. Increase in GP.**

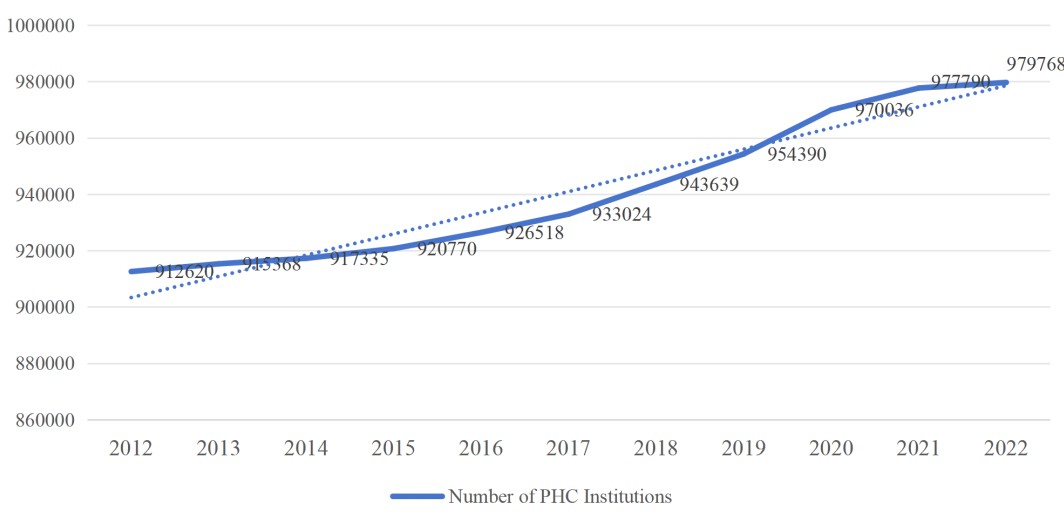

**Fig 2. Increase in PHC.**

missing the underlying root cause of inadequate GP career advancement and mismatches in career trajectories of PHC employees.

Against this background, this study conceptualized GPs as a heterogeneous group and compared straight line with non-straight-line GP as presented in Fig 5 with the aim to assess labor dynamics by comparing two groups within the same field.

Using this unique measure, this study focuses on whether the employment of specialists in the PHC field will cause horizontal or vertical mismatches. Given that SES is a key factor in educational success, it is particularly important to analyze its impact so that the core factors that drive the career advancement of GPs are understood, thereby improving the

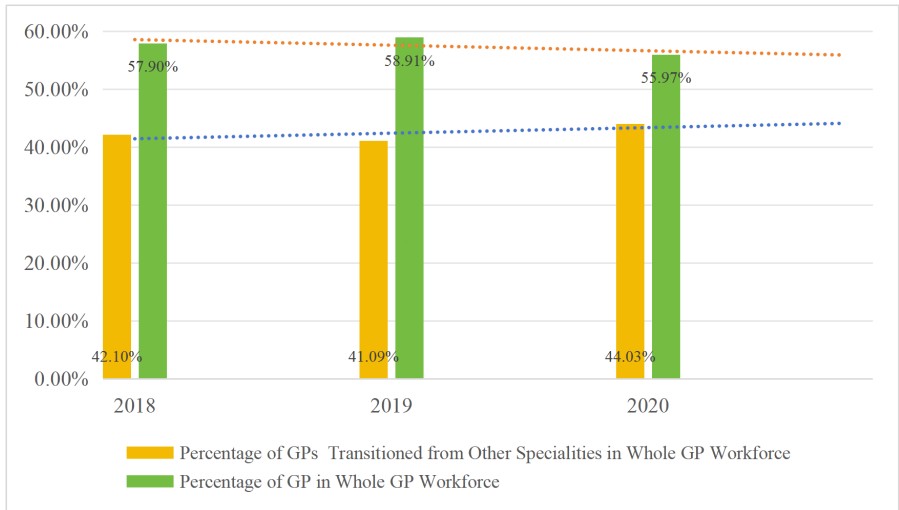

**Fig 3. Increase in SP-GP and Decline in GP in PHC.**

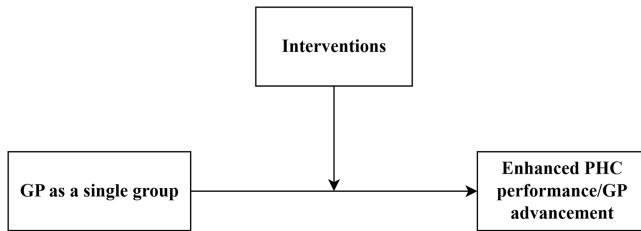

**Fig 4. Conventional path followed by earlier studies that aimed to develop PHC.**

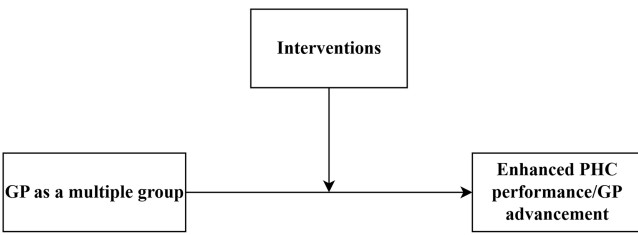

**Fig 5. New dynamic path considered for this study.**

overall sustainability of the PHC system. This is also the core objective of this study. Based on this, the following research questions are proposed:

(1) What are the differences in SES between GP and SP clinical physicians?

(2) What is the ratio of different types of medical graduates employed in the PHC institutions?

(3) Which types of medical graduates achieve better professional success as GPs in PHC centers?

(4) What is the relationship between degree program type, SES and GPs' career advancement?

Notably, a better SES is acknowledged as an issue that significantly influences academic achievement and career development [30]. Phillips et al. [31] and Bodenheimer et al. [32] concluded that hypothesized the following, that:

Ho1: There is no significant difference between SP and GP clinical physicians' career advancement.

Ho2: SES does not mediate the correlation between GPs' degree types and their career advancement.

## Methods

Quantitative and correlational survey design was used for this study. Creswell and Creswell [33] explained that this design can accurately reveal the relationships between variables through statistical analysis.

### Population and sample

The population for this study includes 192 clinical medical graduates in China and the 2018–2019 cohort with 93,000 graduates excluding military medical institutions. In order to obtain the representative sample, multi-stage sampling approach was used [34]. However, other types of medical institutions are: 5-year program+3 SRT (5 years of undergraduate medical education leading to a Bachelor degree+3 years of standardized residency training); 5+3 integration program (5 years of undergraduate education +3 years of a professional Master's program including SRT); and 8-year program (5 years of undergraduate education followed by 3 years of a MD program) [35].

In order to ensure triangulation, this study adopted the proportional sampling method, based on the different types of universities available. Hence, one 8-year, two 5+3, and ten 5-year clinical program colleges were sampled, totaling 13 colleges with 12,767 graduates. However, the total population of interest for this study includes all the 5,946 out of the 12,767 graduates in the 2018–2019 cohort from the 13 medical colleges who work as clinical physicians. The remaining 6,821 either work in the pharmaceutical industry, or work as medical researchers and educators, etc. The entire population of interest (5,946) is considered as the sample, not just to avoid sample bias but provide a random visualization of the graduates' degree type, SES and career advancement (see Table 1 below).

### Data collection and ethical approval

The data collection phase proceeded between June 20th and August 25th, 2025. The data was collected from medical graduates' records held by their undergraduate colleges and the Health Commission. Before the study began, one of the authors visited the selected medical schools/universities and related Health Commission to explain the purpose of

Table 1. Population and sample.

| Program | No. of medical schools | Matriculants/ year (1000s) | No. of graduates (sampled medical schools) | Specialists | | Transferred GP from Specialist (PHC) | General Practitioners | |
|---|---|---|---|---|---|---|---|---|
| | | | | Hospital | PHC | SP- GP | Hospital | PHC |
| 5-year programme | 151a (10 sample selected using ratio model) | 87± | 12156 | 3647 | 911 | 558 | 103 | 285 |
| 5+3 integration programme | 27b (2 sample selected using ratio model) | 4.9± | 376 | 289 | – | 15 | – | – |
| 8-year programme | 14c (1sample selected using ratio model) | 1.1± | 235 | 138 | – | – | – | – |
| Total | 192⇔13 | 93± | 12767 | 4074 | 911 | 573 | 103 | 285 |

the study and apply for permission to conduct the research. All sampled universities and the Health Commission were informed that participation in the study was entirely voluntary and that participants had the right to withdraw at any time. After obtaining approval from the university's Ethics for Research Involving Human Subject Committee with Reference No.: 2025 GMU Ethic Review No. (012), the researchers further explained the required information to each medical school in detail. Subsequently, they collected the data in compliance with the ethics approved.

Demographic characteristics, academic progress, and career advancement information of graduates were provided by the Health Commission. To maintain the anonymity of participants, the sample was sequentially named using a combination of numbers and letters, with the numbering starting from the first case in the list. Meanwhile, the SES data for medical graduates were obtained from secondary records in the Student Financial Aid Office archives of the sampled universities. Secondary data are generally considered more reliable than self-reported data and can effectively reduce potential bias [36]. It should be emphasized that the participating universities did not provide the research team with any declarative documents or identifiable personal information of the subjects during the study.

After the data collection, the research team screened and classified the samples according to the medical education path (GP straight line and non-straight line), and the specific classification criteria are shown in Table 2. Graduates who have received clinical medical education and completed a GP residency training and/or a Master's program in general practice are classified as Full GPs. Meanwhile, those who have received other specialty training (in different fields) after completing clinical medical education and later transferred to general practice positions are defined as SP-GPs with corresponding specialty backgrounds. In addition, graduates who completed clinical medical education and continued to receive other specialist training were classified as Full SP, and the specific classification criteria are shown in Table 2. Thus, a combination of convenience sampling served to select four types of SP-GP because they are the most common types of research samples in most PHCs.

### Measures

The data collected in this study included demographic characteristics, SES, degree type, and career development-related information and the data belonging to the same graduates.

**Control variables**: The demographic data of gender, race, age, and location of residence were included in this study based on their potential impact on career development.

**SES Measurement**: SES was measured using a composite score based on parental education, parental occupation, and household income. Based on this composite score, subjects were classified into four SES levels: low, low-middle, high-middle, and high [37].

**Table 2. Sample grouping.**

| Group name | Years in medical college | Years of clinical rotation | 1-year GP transfer training | Degrees |
|---|---|---|---|---|
| Full general practitioners (GP) | 5 | 3 | No | Bachelor/Master |
| Full Specialist (SP) | 5 | 3 | No | Bachelor/Master |
| Full Specialist (SP) | 8 | 1-2 | No | Doctor |
| Internal Medicine Specialist-GP (IM SP-GP) | 5 | 3 | Yes | Bachelor/Master |
| Emergency Medicine Specialist-GP (EM SP-GP) | 5 | 3 | Yes | Bachelor/Master |
| Pediatrics Specialist-GP (P SP-GP) | 5 | 3 | Yes | Bachelor/Master |
| Obstetrics and Gynaecology Specialist-GP (OB/GYN SP-GP) | 5 | 3 | Yes | Bachelor/Master |

**Career advancement**: Career advancement was assessed using a composite measure of promotion score (vertical promotion) and scope of practice expansion (horizontal expansion) [23–25]. In China, the Medical Professional Title Qualification Examination is held annually to determine the promotion of PHC physicians' professional titles, and the administrative position promotion is also regarded as another type of vertical promotion. Meanwhile, changes in the role of employment or adjustments in daily work tasks can also reflect the expansion of the working responsibilities of graduates. Examples of this are outpatient service and other public prevention work. By comprehensively examining these two dimensions – professional title/ administrative position promotion and responsibility expansion – this study can more comprehensively reveal the differences in the career advancement paths of Full GPs and different types of SP-GPs (see Table 3 below).

### Reliability

This study is based on the analysis of archival data. Methodological strategies such as data quality control, contextual interpretation of data, and transparent or fully disclosed research processes are implemented to ensure the validity and reliability of secondary data [38].

### Data analysis

For RQ1 and RQ2, descriptive analysis with percentages and ratios were used to indicate the SES and type of degree dynamics of PHC employees, respectively. Meanwhile, multiple linear and group regression was employed to address RQ3 and highlight the differences in the career advancement of various types of degree program (horizontal, vertical and career advancement patterns). Finally, mediation analysis used the bootstrap option served to answer RQ4 as presented in Table 4.

**Table 3. Variables and domains.**

| Variables | Domains |
|---|---|
| SES | Low SES = 0; Lower-middle SES = 1<br>Higher-middle SES = 2; High SES = 3 |
| Career advancement | No promotion + no responsibility expansion = 0,<br>no promotion + responsibility expansion = 1,<br>promoted + no responsibility expansion = 2,<br>promoted + responsibility expansion = 3 |
| Type of degree program | FULL GP = 5; Internal Medicine SP-GP = 4; Obstetrics and Gynaecology SP-GP = 3; Emergency Medicine SP-GP = 2; Pediatrics SP-GP = 1, Full SP = 0 |

**Table 4. Research questions and statistical tools.**

| Research questions | Auxiliary tools | Method |
|---|---|---|
| RQ1: What are the differences in SES between GP and SP clinical physicians? | Descriptive analysis, SPSS version 29 | Quantitative |
| RQ2: What is the ratio of different types of medical graduates employed in the PHC institutions? | Descriptive analysis, SPSS version 29 | Quantitative |
| RQ3: Which types of medical graduates achieve better professional success as GPs in PHC centers? | Regression analysis, SPSS version 29 | Quantitative |
| RQ4: What is the relationship between degree program type, SES and GPs' career advancement? | Mediation analysis, Mplus | Quantitative |

## Ethical considerations

This study was approved by a university's ethics committee before it was conducted. Hence, to protect the anonymity of participants, we assigned codes to all participants in numerical order, starting with the first sample. Therefore, the respondents in the samples collected from the universities were numbered 1, 2, 3, 4, etc. In this way, the personal information of all respondents remained anonymous, and the researchers could not know their true identity.

In the process of data collection, we used letters such as A, B, and C in the alphabet to name the universities. Therefore, the respondent codes of the first university (i.e., the university from which we initially collected information) were A1, A2, A3, etc., until the last respondent. The second university was named with letters such as B1, B2, etc., and all documents collected from the universities were treated with pseudonyms.

## Results

### Demographic data of the respondents

Among the 5,946 medical graduates who are of interest to this study, there are slightly more male than female graduates. Regarding ethnicity, Han graduates significantly out-numbered ethnic minorities. Furthermore, there are 19% more graduates from urban areas compared to those from rural areas as illustrated in Table 5.

### Differences in SES between GP and SP clinical physicians

The data illustrates that the majority of SPs (94.30%) are high and higher middle SES, with only 5.70% as lower middle and low SES. Meanwhile, among the GPs, 7.30% are high SES, 17.00% are higher middle SES, 51.70% are lower middle SES and 24.00% are low SES as presented in Fig 6.

### Ratio of different types of medical graduates employed in PHC Institutions

The 1,769 employees working in PHC institutions were categorized based on the earlier grouping of the different types of graduates and the analysis revealed there are 911 (51.50%) Full SP, 285 (16.11%) are Full GP, and 573 are SP-GP (32.39%)—170 (9.61%) Internal Medicine -SP-GP, 155 (8.76%) Obstetrics and Gynaecology -SP-GP, 118 (6.67%) Emergency Medicine -SP-GP and 130 (7.35%) Pediatrics-SP-GP (based on the 4 selected SP-GP groups). The data is presented in Fig 7.

The results strongly suggest that the number of Full SP (51.50%) is almost twice as large as the proportion of SP-GP (32.39%), while the proportion of SP-GP is almost double that of the proportion of Full GP (16.11%), revealing a ratio of 3:2:1. The workforce composition ratio shows that there is a horizontal mismatch problem PHC institutions; that is, specialist physicians (SPs) undertake the positions that should have been held by GPs.

**Table 5. Demographic data.**

| Control Variables | (%) Frequency |
|---|---|
| *Gender* | |
| Male | 57.0 (3389) |
| Female | 43.0 (2557) |
| *Ethnicity* | |
| Han | 87.8 (5221) |
| Minority | 12.2 (725) |
| *Residential location* Urban | 59.5 (3538) |
| Rural | 40.5 (2408) |

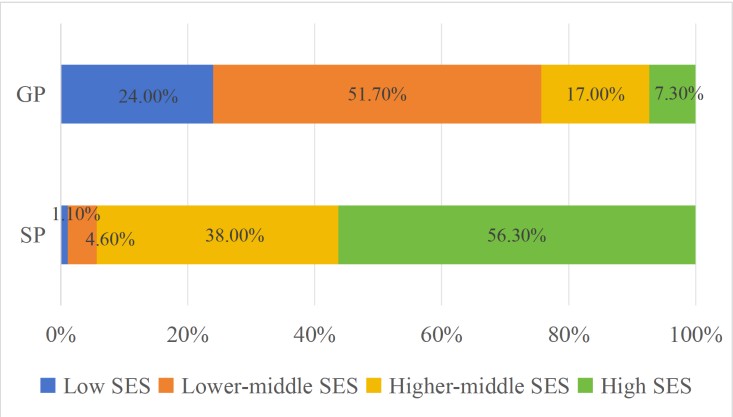

**Fig 6. Differences in SES of clinical physician (SP and GP).**

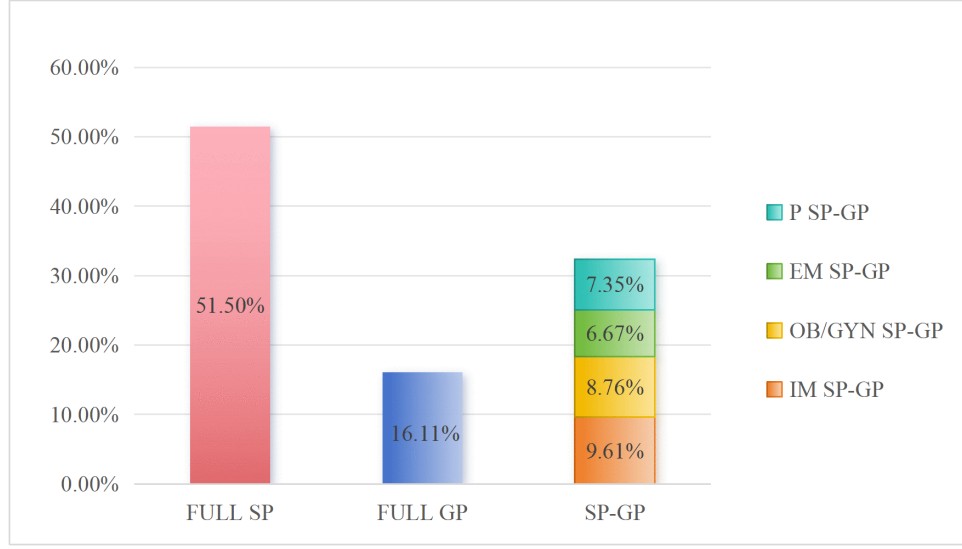

**Fig 7. Ratio of different types of medical graduates employed in the PHCs.**

### Types of medical graduates and better professional success among GPs in PHC centers

Multiple linear regression was used to indicate the significance level and the results revealed that after controlling for gender, ethnicity, age and residential location, compared with Full GP, the SP-GP categories (Internal Medicine SP-GP, Obstetrics and Gynaecology SP-GP, Emergency Medicine SP-GP and Pediatrics SP-GP) reported significantly higher levels of vertical success with standardized coefficients β = 0.306, 0.350, 0.339 and 0.503, respectively. In contrast, the Full GP group demonstrated significantly greater horizontal success when compared with Internal Medicine SP-GP (β = −0.231, P < 0.001), Obstetrics and Gynaecology SP-GP (β = −0.159, P < 0.001), Emergency Medicine SP-GP (−0.115, P < 0.01), Pediatrics-SP-GP (β = −0.151, P < 0.001) as presented in Table 6.

After combining the career advancement dimensions and using group regression analysis, the results revealed that compared with the reference group of Pediatrics-SP-GPs, Full GPs (β = −0.864, p < 0.001) exhibited the lowest career

**Table 6. Impact of degree program type on vertical and horizontal career advancement.**

| Predictor | Vertical advancement | | Horizontal advancement | |
|---|---|---|---|---|
| | β | SE | β | SE |
| Gender | .030(1.009) | .036 | −.005(−.139) | .037 |
| Ethnicity | .039(1.282) | .049 | −.011(.316) | .052 |
| Age | .046(1.560) | .015 | −.011(−.331) | .016 |
| Residential location | .007(.224) | .037 | .006(−.169) | .038 |
| Internal Medicine SP-GP | .306***(9.084) | .049 | −.231***(−6.044) | .051 |
| Obstetrics and Gynaecology SP-GP | .350 ***(10.264) | .052 | −.159 ***(−4.095) | .054 |
| Emergency Medicine SP-GP | .339 ***(10.394) | .055 | −.115**(−3.111) | .058 |
| Pediatrics-SP-GP | .503 ***(15.311) | .053 | −.151***(−4.042) | .056 |
| R² | .263 | | .050 | |
| Adjusted R² | .256 | | .041 | |

**Note**: Ref.=Full GP. ***$P<.001$, **$P<.01$, *$P<.05$.

advancement. This was followed by Obstetrics and Gynaecology SP-GP (β=−0.220, p<0.05) and Internal Medicine SP-GP (β=−0.211,p<0.05). Conversely, the career advancement of the Emergency Medicine SP-GP group did not differ from the Pediatrics-SP-GP group significantly (β=−0.005, p>0.05). As such, $H_O1$ is rejected (see Table 7).

## Relationship between degree program type, SES and GPs' career advancement

Initially, descriptive statistics were used to analyze the SES dynamics of the various SP-GPs and Full GPs employed in the PHC system. The data revealed that 22.46% of Full GPs are low SES while 67.72% are low-middle SES and only 9.47% are from higher-middle SES. However, the majority of SP-GPs (43.11%) are from higher-middle SES, 19.37% are from high SES, and only 36.82% and 0.70% are from lower-middle and low SES, respectively (see Fig 8).

Moreover, the results indicate that all the heterogenous SP-GP groups rarely have low SES. Specifically, among IM SP-GP group, 35.29% are lower-middle SES, 54.71% higher-middle SES, and 8.82% high SES. The SES distribution of Obstetrics and Gynaecology SP-GP almost mirrors that of the Internal Medicine SP-GP group, while with 20.65% high SES. Emergency Medicine SP-GP shows 33.05% of high-SES, the highest across all SP-GP categories. Referring to

**Table 7. Impact of degree program type on career advancement.**

| Variables | β | SE | t | P | 95% CI | |
|---|---|---|---|---|---|---|
| | | | | | Lower Bound | Upper Bound |
| Gender-female | −.063 | .064 | −.973 | .331 | −.189 | .064 |
| Age | .032 | .028 | 1.153 | .249 | −.023 | .087 |
| Residential location-rural | −.108 | .066 | −1.638 | .102 | −.238 | .022 |
| Ethnicity-minority | −.108 | .066 | −1.638 | .102 | −.238 | .022 |
| GP type : P SP-GP as reference | | | | | | |
| FULL GP | −.864 | .096 | −8.972 | .000*** | −1.053 | −.675 |
| IM SP-GP | −.211 | .107 | −1.983 | .048* | −.421 | −.002 |
| OB/GYN SP-GP | −.220 | .110 | −1.990 | .047* | −.436 | −.003 |
| EM SP-GP | −.005 | .116 | −.041 | .967 | −.232 | .222 |

**Note**: ***$P<.001$, **$P<.01$, *$P<.05$

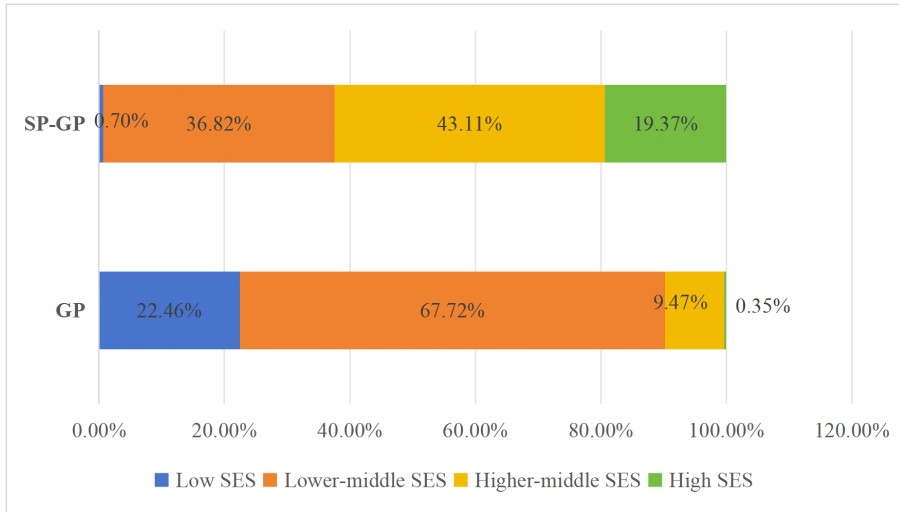

**Fig 8. SES dynamics of GP and SP-GP in PHC institutions.**

Pediatrics SP-GP, 27.69% are from higher-middle SES, 53.08% are lower-middle SES, 19.23% are high SES, as illustrated in Fig 9. This implies that regardless of SP-GP type, all SP-GP groups enjoy better SES compared to full GPs.

Furthermore, mediation analysis was conducted and the analysis indicated a significant correlation between SES and career advancement was evident ($\beta = 0.133$, $P < 0.001$). The direct effect revealed there was a correlation between degree program type and career advancement among the various GPs working in PHC institutions using the Full GP group as a reference; Internal Medicine SP-GP$\rightarrow$CA ($\beta = 0.046$, $P < 0.05$), Obstetrics and Gynaecology SP-GP$\rightarrow$CA ($\beta = 0.691$, $P < 0.001$), Emergency Medicine SP-GP$\rightarrow$CA ($\beta = 0.565$, $P < 0.001$), Pediatrics SP-GP$\rightarrow$CA ($\beta = 0.119$, $P < 0.001$) as detailed in Table 8.

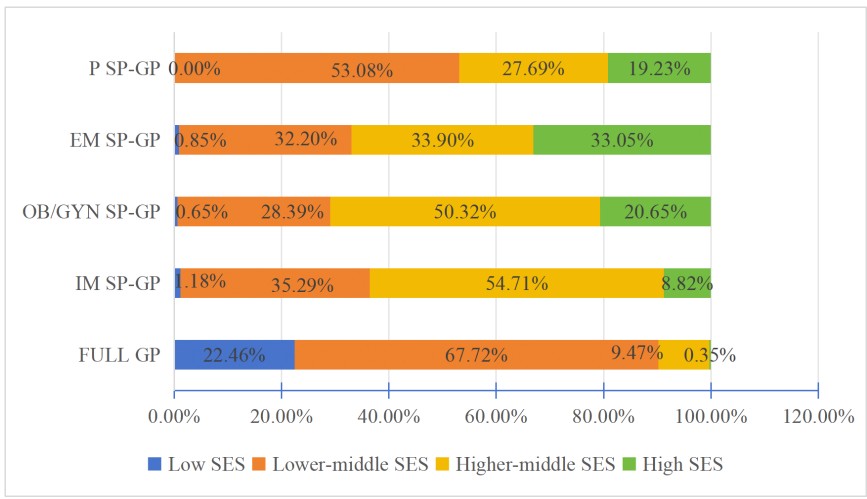

**Fig 9. SES distribution of employees working as GP in PHC institutions.**

**Table 8. Mediation of SES on degree program type and career advancement.**

| Path/effect | Estimate | SE | P | 95% CI | |
|---|---|---|---|---|---|
| **Reference = Full GP** | | | | Lower boundary | Upper boundary |
| Direct effect | | | | | |
| **SES→CA** | .133 | .027 | .000*** | .066 | .155 |
| IM SP-GP→CA | .046 | .022 | .039* | .002 | .060 |
| OB/GYN SP-GP→CA | .691 | .020 | .000*** | 1.086 | 1.239 |
| EM SP-GP→CA | .565 | .015 | .000*** | .974 | 1.090 |
| P SP-GP→CA | .119 | .021 | .000*** | .194 | .392 |
| Total effect | | | | | |
| IM SP-GP→CA | .086 | .020 | .000*** | .030 | .085 |
| OB/GYN SP-GP→CA | .717 | .019 | .000*** | 1.134 | 1.280 |
| EM SP-GP→CA | .585 | .015 | .000*** | 1.010 | 1.128 |
| P SP-GP→CA | .144 | .019 | .000*** | .266 | .439 |
| Total indirect effect: GP Type→SES→CA | | | | | |
| IM SP-GP→SES→CA | .040 | .009 | .000*** | .016 | .039 |
| OB/GYN SP-GP→SES→CA | .026 | .007 | .000*** | .025 | .068 |
| EM SP-GP→SES→CA | .021 | .005 | .000*** | .021 | .060 |
| P SP-GP→SES→CA | .024 | .006 | .000*** | .035 | .090 |

**Note:** ***$P < .001$, **$P < .01$, *$P < .05$.

Finally, SES significantly mediates the relationship between GPs' degree program type and their career advancement. The specific indirect paths coefficients are as follows: Internal Medicine SP-GP→SES→CA ($\beta = 0.040$, $P < 0.001$), Obstetrics and Gynaecology SP-GP→SES→CA ($\beta = 0.026$, $P < 0.001$), Emergency Medicine SP-GP→SES→CA ($\beta = 0.021$, $P < 0.001$), Pediatrics SP-GP→SES→CA ($\beta = 0.024$, $P < 0.001$), detailed in Table 8. Hence, $H_O2$ is rejected.

## Discussion

Studies by Wu et al. [39] and Deutch [40] explained that most specialists come from higher SES backgrounds. Likewise, studies by Fujii et al. [41] and Xu et al. [20] revealed that most GPs originate from lower SES origins. Similarly, in China, the SES of an individual determines the type of university and even the program a student can apply for. It is generally the case that low SES students can only access normal universities which offer 5-year and 5 + 3 Bachelor degree programs while children of the elite can access the 8 year program. Moreover, because GPs generally work in remote areas or at the grassroots level, most GPs are from the low SES which is more accessible to them for internships and jobs, while the other specialists program which are more popular and competitive are mostly accessed by the high SES [42]. Additionally, McMahon and Cunningham [43] indicated that most full GPs rarely apply for further education or professional development because they cannot afford the financial costs. This further highlights how SES controls or influences the education an individual can obtain and, in this situation, poses a challenge to the sustainable development and service provisions of PHC institutions.

Similarly, the research has highlighted that in the GP workforce, a large number of practitioners transitioned from other specialties [11]. Although, ideally, GP positions are meant to be occupied by full GPs who have a diversified skill and are professionally more capable of handling wider responsibilities, due to preferential policies, the SP-GPs are more than the Full GPs in Chinese PHCs. This subsequently limits the capacity of PHC and results in horizontal mismatch. This situation disrupts the structural integrity of the medical system and may lead to the deprivation of opportunities for the Full GP.

Zhang et al. [44] found that SP graduates experience greater success compared to their GP counterparts. This motivates many students to choose specialist fields instead of becoming a GP, and attracts many high SES students to the program [45]. Similarly, this study reported that SPs have better career advancement opportunities compared to the GPs. The finding shows that SPs have better vertical advancement (promotions) while GPs have higher horizontal career advancement, which basically refers to more responsibilities and daily tasks. As such, GPs due to their diversified skills have to do more work as they advance in their career and may have less promotion opportunities in China. However, this finding contradicts the work done by Deutsch et al. [40], who noted that many graduates join the GP ranks due to the success attached to the profession. As opined by Velgan et al. [46], many GPs migrate to more urban centers or seek employment in Western countries, thereby weakening the PHC system.

In this circumstance, whether the specialists perform better than the Full GP in the PHC, it is a case that is causing much concern. Questions will still arise on whether medical education colleges are not producing competent Full GPs, and whether the transfer of SP into GP does not result in huge investment waste (if the SP will transfer to GP, it is not feasible to waste resources training SP). Will this further result in weakening the specialist practitioners and ultimately lead to a shortage of real required specialists?

The findings of this study highlight the dominance of SES in shaping medical graduates' career advancement which supports international discourse on SES, academic achievement and students' career choice and advancement. SES therefore requires a cohesive and integrative approach to reduce its effect on individuals from low SES backgrounds. Therefore, policies pertaining to medical education and health administration may not be creating a comprehensive healthcare system for society at large; instead, they re-enforce SES effects on education.

### Limitations and further research

Despite this study employing a unique technique for the purposes of analysis, it has some limitations. Firstly, this research is limited in its dependence on quantitative analysis. Further studies should use qualitative analysis to provide in-depth investigations on why the situation appears the way it is. Secondly, it is suggested that further research should use primary data from a larger cohort of respondents. Lastly, this study mainly focuses on the graduates who subsequently choose to work as clinicians among clinical medical graduates, and explores the relationship between SES of graduates who choose to work in different clinical specialties and their subsequent career advancement. Because graduates who do not choose to continue to work as clinicians cannot compare their different clinical specialty choices and career development, they are not included. Further studies will try to expand the sample scope to the whole clinical medical graduates to enrich the findings of this study.

### Conclusions

This current research indicates that Full SPs are twice the number of SP-GPs, while SP-GP are double the proportion of Full GPs in PHC institutions. Interestingly, SP-GPs experience more vertical career advancement while full GPs experience more horizontal advancement. Additionally, SES mediates the association between degree program type and career advancement. Unfortunately, as SP-GPs increasingly work at PHC centers, they further deprive the career advancement opportunities of GPs, and this results in horizontal mismatch at a larger end. In effect it hinders the required skills for the sustainable development of PHC centers. This study highlights that the inequality and mismatch among medical graduates and their jobs further exacerbate the challenges of the PHC system. There is subsequently a need for a rigorous and effective reform in medical education and work dynamics of the GPs. This means addressing the root problem – SES. Moreover, more resources and support should be allocated to PHC institutions and GPs.

### Supporting information

**S1_File. Data for** Table 6**: Influence of GP degree program type on vertical and horizontal career advancement.** (PDF)

**S2_File. Data for** Table 7**: Influence of degree program type on career advancement.**
(PDF)

**S3_ File. Data for** Table 8**: Medicating effect of SES on degree program type and career advancement.**
(PDF)

## Author contributions

**Conceptualization:** Gazi Mahabubul Alam.

**Data curation:** Dandan Zheng, Gazi Mahabubul Alam, Karima Bashir, Miao Lei.

**Formal analysis:** Dandan Zheng, Gazi Mahabubul Alam, Karima Bashir.

**Funding acquisition:** Dandan Zheng.

**Investigation:** Dandan Zheng, Gazi Mahabubul Alam, Karima Bashir.

**Methodology:** Gazi Mahabubul Alam.

**Project administration:** Gazi Mahabubul Alam.

**Resources:** Dandan Zheng, Gazi Mahabubul Alam, Miao Lei.

**Software:** Dandan Zheng.

**Supervision:** Gazi Mahabubul Alam, Karima Bashir, Norlizah Che Hassan.

**Validation:** Gazi Mahabubul Alam.

**Visualization:** Karima Bashir.

**Writing – original draft:** Dandan Zheng, Gazi Mahabubul Alam, Karima Bashir, Miao Lei.

**Writing – review & editing:** Dandan Zheng, Gazi Mahabubul Alam, Karima Bashir, Miao Lei.

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
