## [Decision Letter · Decision Letter 0]

18 Dec 2025

PONE-D-25-58852Does a degree in medicine or a specialist programme or socioeconomic status advance the career of general practitioners in primary healthcare? — a comparison of five types of graduatesPLOS One?

Dear Dr. Alam,

Thank you for submitting your manuscript to PLOS ONE. After careful consideration, we feel that it has merit but does not fully meet PLOS ONE’s publication criteria as it currently stands. Therefore, we invite you to submit a revised version of the manuscript that addresses the points raised during the review process.

Please note that you will be asked several questions and that an extensive revision of your English writing is required, preferably by a native speaker. The new version must be thoroughly revised, and all changes must be indicated in the revision letter.

We look forward to receiving your revised manuscript.

Kind regards,

Ricardo Q. Gurgel, PhD

Academic Editor

PLOS One

Journal Requirements:

“This research was supported by the Philosophy and Social Sciences Planning Project of Guizhou Province, 2023 (Project No. 23GZQN73)”

3. In the online submission form, you indicated that “Further access to the data is available upon reasonable request from the first author (ZDD).”

4. Please ensure that you refer to Figure 3 in your text as, if accepted, production will need this reference to link the reader to the figure.

5. We note you have included a table to which you do not refer in the text of your manuscript. Please ensure that you refer to Table 2 in your text; if accepted, production will need this reference to link the reader to the Table.

Reviewers' comments:

Reviewer's Responses to Questions

**Comments to the Author**

1. Is the manuscript technically sound, and do the data support the conclusions?

Reviewer #1: Yes

Reviewer #2: Partly

2. Has the statistical analysis been performed appropriately and rigorously?

Reviewer #1: Yes

Reviewer #2: I Don't Know

3. Have the authors made all data underlying the findings in their manuscript fully available?

Reviewer #1: Yes

Reviewer #2: No

4. Is the manuscript presented in an intelligible fashion and written in standard English?

Reviewer #1: Yes

Reviewer #2: No

Reviewer #1: Congratulations to the authors. The manuscript has an interesting and current theme, the relationship between socioeconomic level and professional success among doctors working in primary health care in China. On the one hand, obstacles to the choice of graduates to work in primary health care were identified, and, on the other hand, the migration of specialists to this area. The main obstacle is the doctor's socioeconomic level.

The manuscript uses a population of 12,696 medical graduates from 13 universities, all located in the southwest region of China. Including only clinical physicians, the sample consisted of 1,753 physicians. NoThe text is well written, with well-described method and statistical analysis, as well as results and discussion. However, it is observed that the introduction is excessively long, with 8 pages excluding the figures.It is suggested that the authors summarize the introduction, because, in the case of a manuscript that uses qualitative methodology, an objective text is the best choice.

In addition, the text makes quite much use of abbreviations, which makes reading difficult. It is suggested that the abbreviations for the 5 types of courses, such as OB/GYN SP-GP, be replaced by one or two words (such as "academic master's degree" for example), properly described in the method, starting from the chart on page 12.

Reviewer #2: The authors have done an analysis of 5 types of graduates from medical training programs in China and have found that low socioeconomic status is associated with graduates becoming general practitioners as compared to specialists. Unfortunately, the manuscript as written is not acceptable for publication. The background is overly long, makes provocative claims that are not supported by data in the references that are cited, and does not sufficiently put the research question into proper context. The authors should consider the worldwide readership of the journal, and consider a background on the Chinese medical training system that helps put the research that was undertaken into context.

In terms of methods, the authors utilized a "Career Advancement" composite score of promotion scores and responsibility expansion. If this is a validated composite score, a reference should be cited, but otherwise, there should be much more transparency about how this score is defined. It's also unclear if this career advancement score is informative- it does not take into account the physician/practitioner's interests and inclinations. In the methods (which is also unusual as it should be part of the results), it is noted that the population of interest was 5,893 out of 12,696 graduates in the 2018-2019 cohort from 13 medical colleges. It's noted there that the remaining 6803 trainees did not go onto provide patient care (went to pharmaceutical industry, work as medical researchers, educators). Surely, this significant attrition should be discussed as a confounding factor as I could not find analysis of the SES status of those who did not go on to practice clinically.

While these findings are interesting and support continued data that SES status affects choice of medical training, it is unclear if this work adds to what is already known. There are data that SES may impair students from being accepted into programs in various countries- is this a problem in China as well, and are GP programs less competitive than GP-SP or specialist programs? This is not really evaluated or discussed in this manuscript and instead some rather bold conclusions are posited that are not completely supported by the data in my opinion.

**Do you want your identity to be public for this peer review?** For information about this choice, including consent withdrawal, please see our For information about this choice, including consent withdrawal, please see our Privacy Policy .

Reviewer #1: **Yes:** Rosana CipolottiRosana Cipolotti

Reviewer #2: No

---

## [Author Response · Author response to Decision Letter 1]

12 Jan 2026

4th January, 2026

Professor Ricardo Q. Gurgel, PhD

The Academic Editor

PLOS One

Subject: Submission of the revised manuscript

Dear Ricardo Q. Gurgel, PhD,

We humbly remain grateful to you for appraising our paper and providing us with insightful comments about our study. Two esteemed reviewers have reviewed the paper, and we remain very grateful to their meticulous inputs that are fundamental for enhancing this paper. We have addressed your comments and those from the two reviewers and they are reported in blue color text. Kindly find below the responses for each comment, and we remain committed to ensuring we satisfy your requirements for publication.

We appreciate your time and consideration.

Thank you,

Authors

Response to Editorial Comments

Comment 1

Response:

We sincerely appreciate your comment. We have carefully reviewed PLOS ONE’s style requirements and ensured that the revised manuscript complies with them, including those for file namings. See entire manuscript and attched files. Thank you.

Comment 2

Thank you for stating the following financial disclosure:

“This research was supported by the Philosophy and Social Sciences Planning Project of Guizhou Province, 2023 (Project No. 23GZQN73)”

Response:

Thank you for the kind clarification. Indeed, the funding bodies had no involvement in the conduct of this study. The Funder has no role and the statement: “The funders had no role in study design, data collection and analysis, decision to publish, or preparation of the manuscript.” have been amended as needed and we appreciate your assistance in updating the online submission form accordingly. Thank you.

Comment 3

In the online submission form, you indicated that “Further access to the data is available upon reasonable request from the first author (ZDD).”

Response:

In accordance with PLOS data availability requirements, we have revised the Availability of Data and Materials section of the manuscript (see Page 20, Line 442-443). All data underlying the findings are now provided as Supplementary Files accompanying the manuscript and are freely available to other researchers. See supplementary file 11,12 and 13. Thank you.

Comment 4

Please ensure that you refer to Figure 3 in your text as, if accepted, production will need this reference to link the reader to the figure.

Response:

Thank you for the reminder. We have revised the manuscript to explicitly refer to Figure 3 in the main text at the relevant location, ensuring proper linkage between the text and the figure. See page 5, line148. Thank you

Comment 5

We note you have included a table to which you do not refer in the text of your manuscript. Please ensure that you refer to Table 2 in your text; if accepted, production will need this reference to link the reader to the Table.

Response:

We have reviewed the manuscript carefully and added an explicit in-text reference to Table 2 in the relevant section of the text to ensure that readers are properly directed to the table and that production requirements are met. See page 9, line230. Thank you.

Comment 6

Response:

After reviewing all reviewer comments, we note that the reviewers did not recommend citing any specific previously published works. As such, no changes to the reference list were required in this regard.

Reviewer 1

Comment 1

Congratulations to the authors. The manuscript has an interesting and current theme, the relationship between socioeconomic level and professional success among doctors working in primary health care in China. On the one hand, obstacles to the choice of graduates to work in primary health care were identified, and, on the other hand, the migration of specialists to this area. The main obstacle is the doctor's socioeconomic level.

Response:

Thank you for your kind and generous comments. We are sincerely grateful to you for your constructive criticism that has helped to improve our paper. We hope to fulfill your requirements and hope that this manuscript will be accepted for publication.

Comment 2

The text is well written, with well-described method and statistical analysis, as well as results and discussion. However, it is observed that the introduction is excessively long, with 8 pages excluding the figures. It is suggested that the authors summarize the introduction, because, in the case of a manuscript that uses qualitative methodology, an objective text is the best choice.

Response:

Thank you for your positive and constructive comments. We sincerely appreciate your recognition of the writing quality, methodology, statistical analysis, results, and discussion. We have summarized the Introduction. Please see page 2, lines 56–72. Thank you.

Comment 3

In addition, the text makes quite much use of abbreviations, which makes reading difficult. It is suggested that the abbreviations for the 5 types of courses, such as OB/GYN SP-GP, be replaced by one or two words (such as "academic master's degree" for example), properly described in the method, starting from the chart on page 12.

Response:

Thank you for this helpful suggestion. We have revised the manuscript to ensure that all abbreviations are spelled out in full. Please see from page 9, line 232-233.

Response to Reviewer 2 comments

Comment 1

The authors have done an analysis of 5 types of graduates from medical training programs in China and have found that low socioeconomic status is associated with graduates becoming general practitioners as compared to specialists. Unfortunately, the manuscript as written is not acceptable for publication. The background is overly long, makes provocative claims that are not supported by data in the references that are cited, and does not sufficiently put the research question into proper context. The authors should consider the worldwide readership of the journal, and consider a background on the Chinese medical training system that helps put the research that was undertaken into context.

Response:

Thank you for the insightful comments provided. We have revised and summarised the background section. Likewise, the references we have cited to support our claims are journals recognized by web of science. Additionally, we have provided a background on the Chinese medical training system in order to highlight the research context for worldwide readership. Thank you. See page 2-6, lines 56–175.

Comment 2

In terms of methods, the authors utilized a "Career Advancement" composite score of promotion scores and responsibility expansion. If this is a validated composite score, a reference should be cited, but otherwise, there should be much more transparency about how this score is defined. It's also unclear if this career advancement score is informative- it does not take into account the physician/practitioner's interests and inclinations. In the methods (which is also unusual as it should be part of the results), it is noted that the population of interest was 5,893 out of 12,696 graduates in the 2018-2019 cohort from 13 medical colleges. It's noted there that the remaining 6803 trainees did not go onto provide patient care (went to pharmaceutical industry, work as medical researchers, educators). Surely, this significant attrition should be discussed as a confounding factor as I could not find analysis of the SES status of those who did not go on to practice clinically.

Response: Thank you very much for your insightful and constructive comments. Regarding the measurement of career advancement, we have added relevant literature to support the validity of this composite measure and have provided a clearer and more detailed explanation of how the score was defined and calculated to improve transparency. See page 10, lines 242–252.

In addition, following your suggestion concerning the structure of the manuscript, we have revised the Methods and Results sections by moving the demographic information of the samples from the Methods section to the Results section. See page 12, line 280-286.

With respect to graduates who are not in clinical practice, they were omitted because they are not part of the sample selected for the study. Moreover, graduates who do not choose to continue to work as clinicians do not have any specialty. However, we agree that this represents an important consideration, so it is discussed as a limitation of the study. Thank you. See page 19, lines 417–423.

Comment 3

While these findings are interesting and support continued data that SES status affects choice of medical training, it is unclear if this work adds to what is already known. There are data that SES may impair students from being accepted into programs in various countries- is this a problem in China as well, and are GP programs less competitive than GP-SP or specialist programs? This is not really evaluated or discussed in this manuscript and instead some rather bold conclusions are posited that are not completely supported by the data in my opinion.

Response: Thank you for your insightful comments. Firstly, regarding whether GP programs are less competitive than GP-SP or specialist programs in China, we have illustrated this issue explicitly in the Discussion section. See page 17, lines 368–375.

Secondly, concerning the contribution of this study beyond existing international evidence, We explicitly compare our results with prior studies from other countries and highlight how the Chinese context both aligns with and differs from these patterns (see Page 18, lines 387–398). Specifically, this study examined various categories of SP-GP to explore whether any differences occur among various SP-GP graduates. These findings have been evaluated and discussed (See Page 18). Thank you.

---

## [Decision Letter · Decision Letter 1]

15 Mar 2026

Does a degree in medicine or a specialist programme or socioeconomic status advance the career of general practitioners in primary healthcare?

PONE-D-25-58852R1

Dear Dr. Alam,

We’re pleased to inform you that your manuscript has been judged scientifically suitable for publication and will be formally accepted for publication once it meets all outstanding technical requirements.

Kind regards,

Ricardo Q. Gurgel, PhD

Academic Editor

PLOS One

Additional Editor Comments (optional):

Reviewers' comments:

Reviewer's Responses to Questions

**Comments to the Author**

Reviewer #1: All comments have been addressed

Reviewer #3: All comments have been addressed

Reviewer #4: All comments have been addressed

Reviewer #5: All comments have been addressed

2. Is the manuscript technically sound, and do the data support the conclusions?

Reviewer #1: Yes

Reviewer #3: Yes

Reviewer #4: Yes

Reviewer #5: Yes

3. Has the statistical analysis been performed appropriately and rigorously?

Reviewer #1: Yes

Reviewer #3: Yes

Reviewer #4: I Don't Know

Reviewer #5: Yes

4. Have the authors made all data underlying the findings in their manuscript fully available?

Reviewer #1: Yes

Reviewer #3: Yes

Reviewer #4: (No Response)

Reviewer #5: Yes

5. Is the manuscript presented in an intelligible fashion and written in standard English?

Reviewer #1: Yes

Reviewer #3: Yes

Reviewer #4: Yes

Reviewer #5: Yes

Reviewer #1: All comments have been addressed by the authors, who reduced some paragraphs on Introduction and they have revised the manuscript and revised all abbreviations, those were spelled out in full

Reviewer #3: I would propose to adjust the abstract. The research objective in the abstract should be stated more explicitly. Some terminology may also not be immediately clear to an international audience, particularly expressions such as “specialists transferred general practitioners (SP-GPs)” and “full general practitioners,” which would benefit from brief clarification in introduction. I suggest to adjust the statement that medical education has become a “money-driven program” - it may appear overly strong and could be framed more cautiously while still emphasizing the role of socioeconomic status in shaping educational pathways and career advancement.

Finally, the manuscript should ensure that all abbreviations used in tables and figures are clearly explained in the corresponding captions or notes so that readers can easily interpret the presented information.

Reviewer #4: This is a timely article, and the authors have addressed the queries/comments raised by the initial reviewers.

Reviewer #5: This article fulfills all the required criteria of PLOS one. The authers' conceptualization of research idea and writing styles, all are provided in a standard way.

**Do you want your identity to be public for this peer review?** For information about this choice, including consent withdrawal, please see our For information about this choice, including consent withdrawal, please see our Privacy Policy .

Reviewer #1: **Yes:** Rosana CipolottiRosana Cipolotti

Reviewer #3: No

Reviewer #4: No

Reviewer #5: **Yes:** RUBINA HANIFRUBINA HANIF

---

## [Editor Report · Acceptance letter]

PONE-D-25-58852R1

PLOS One

Dear Dr. Alam,

I'm pleased to inform you that your manuscript has been deemed suitable for publication in PLOS One. Congratulations! Your manuscript is now being handed over to our production team.

Kind regards,

on behalf of

Professor Ricardo Q. Gurgel

Academic Editor

PLOS One